# EXPLORING IMAGE-TEXT DISCREPANCY FOR UNIVERSAL FAKE IMAGE DETECTION

## ABSTRACT

With the rapid development of generative models, detecting generated images to prevent their malicious use has become a critical issue recently. Existing methods frame this challenge as a binary image classification task. However, such methods focus only on visual space, yielding trained detectors susceptible to overfitting specific image patterns and incapable of generalizing to unseen models. In this paper, we address this issue from a multi-modal perspective and find that fake images exhibit more distinct discrepancies with corresponding captions compared to real images. Upon this observation, we propose to leverage the **I**mage-**T**ext **D**iscrepancy (**TIDY**) in joint visual-language space for *universal fake image detection*. Specifically, we first measure the distance of the images and corresponding captions in the latent spaces of CLIP, and then tune an MLP head to perform the usual detection task. Since there usually exists local artifacts in fake images, we further propose a global-to-local discrepancy scheme that first explores the discrepancy on the whole image and then each semantic object described in the caption, which can explore more fine-grained local semantic clues. Extensive experiments demonstrate the superiority of our method against other state-of-the-art competitors with impressive generalization and robustness on various recent generative models.

## 1 INTRODUCTION

Recent years have witnessed the rapid development of generative models, such as generative adversarial networks (GANs) (Goodfellow et al., 2014; Karras et al., 2018; 2019; Brock et al., 2018; Park et al., 2019; Zhu et al., 2017) and diffusion models (Dhariwal & Nichol, 2021; Nichol et al., 2021; Rombach et al., 2022; Gu et al., 2022; Ramesh et al., 2022). These generative models enable users to create high-quality synthetic images at very low cost. However, this accessibility also presents a double-edged sword, as perpetrators can easily generate fake images for malicious use, such as using Deepfakes [1] to mislead the public, defame celebrities, and even fabricate evidence, leading to severe social, privacy, and security concerns (Suwajanakorn et al., 2017; Devlin & Cheetham, 2023). Therefore, developing general and effective fake image detectors has become a critical issue.

A common approach to tackling the fake image detection problem is to frame it as a binary image classification task, discriminating between real and fake images. Typically, a dataset of real and fake images is used to train a binary classifier (Wang et al., 2020; 2023), but this approach often leads to overfitting on specific image patterns, limiting the model's generalization capability. For instance, some detectors rely on artifacts introduced by specific model architectures (Tan et al., 2024), which constrains their effectiveness to those particular architectures. In contrast, universal detection methods (Ojha et al., 2023) leverage the vision encoder of Contrastive Language-Image Pre-training (CLIP)(Radford et al., 2021) to improve the generalization of visual representations through its zero-shot abilities. However, these methods focus solely on visual cues, neglecting the language component, which is a key driver of CLIP's strong generalization performance.

Given these limitations, we pose the following question: Can we develop a universal fake image detector that generalizes to all generated images, regardless of model architectures or hyperparameters? While existing detectors rely solely on visual clues, we propose to tackle this challenge from a multimodal perspective using pre-trained vision-language models (VLMs). We first investigate

---

[1]https://github.com/deepfakes/faceswap

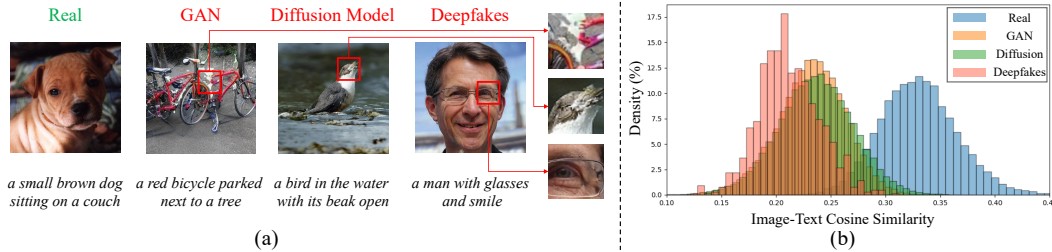

Figure 1: **Motivation behind our method.** We show the local pattern difference between real images and various fake images generated by different models, including GAN, diffusion model, and deepfakes in (a). The cosine similarity between their CLIP's image and text embeddings in (b) shows that fake images exhibit more discrepancy than real images.

some examples of real and different fake images, including GAN, diffusion model, and deepfakes, as shown in Fig. 1 (a). Specifically, we exploit both image patterns and corresponding generated captions. We can observe that different generative models lead to different types of local artifacts, which may cause visual-only detectors cannot generalize well. By incorporating cues from both modalities and leveraging their semantic relationships, a detector would be less prone to overfitting on low-level, visual-only patterns, thus avoiding overfitting to training images. To explore this, we further examine real and fake images in the joint visual-language space by calculating the cosine similarity between image and caption representations learned from CLIP. For the text space, we use corresponding captions that provide relevant semantic clues to describe the images. As illustrated in Fig. 1 (b), real images exhibit higher image-text similarity compared to various fake images, which can serve as a discriminative clue for detection.

Therefore, we propose to leverage the **I**mage-**T**ext **D**iscrenpanc**y** (**TIDY**) for universal fake image detection. Specifically, we first measure the distance of images and their corresponding captions in the joint vision-language space of a pre-trained CLIP and then tune an MLP head for detection. Considering the semantic divergence of local and global patches (Zhang et al., 2022; Li et al., 2019) and the artifacts in local patches of fake images (Fig. 1 (a)), we further propose a global-to-local discrepancy scheme that mines the discrepancy on both the whole image and each semantic object described in the caption, which could explore more fine-grained local semantic clues and benefit the detection. Our main contributions are summarized as follows:

- We frame the fake image detection task from a multimodal image-text perspective and find that the fake images exhibit more distinct discrepancies with corresponding captions compared to real images in joint vision-language latent space.

- We propose **TIDY** to achieve universal fake image detection by measuring the distance of images and captions in a joint vision-language space of CLIP and then tuning an MLP head for detection. Moreover, a global-to-local discrepancy scheme is introduced to explore more fine-grained local semantic clues.

- Extensive experiments on various generative models demonstrate the superiority of our proposed method against other state-of-the-art competitors with impressive generalization and robustness.

## 2   RELATED WORK

**Fake image detection.** With the rapid development of generative models, such as GAN (Goodfellow et al., 2014; Karras et al., 2018; 2019; Brock et al., 2018; Park et al., 2019; Zhu et al., 2017) and diffusion models (Dhariwal & Nichol, 2021; Nichol et al., 2021; Rombach et al., 2022; Gu et al., 2022; Ramesh et al., 2022), a variety of detectors have been proposed to combat the malicious use of AI-generated fake images. Some methods focus on the visual artifacts or traces left by generative models in fake images, such as the noise residual (Yu et al., 2019), face boundaries (Li et al., 2020), patch-level artifacts (Chai et al., 2020), compression traces (Agarwal & Farid, 2017) and frequency clues (Qian et al., 2020). To train the classifier, other methods design specific representations or augmentations, such as (Wang et al., 2020) where pre- and post-processing with data augmentation

are carefully designed to build a universal GAN detector. To detect diffusion-generated images, DIRE (Wang et al., 2023) introduces reconstruction error. To boost generalization, recent methods have exploited pretrained models , such as UniFD (Ojha et al., 2023) that utilizes a pre-trained CLIP-ViT (Radford et al., 2021) model to learn the general image representation for detection.

These methods, however, focus only on the difference on low-level visual image patterns, which may lead to limited generalization on unseen generative models. Although some existing methods tried to use the vision language models (VLMs), such as UniFD (Ojha et al., 2023), they also explored only in visual space. Whereas, we find that there exists significant discrepancy between the images and corresponding captions in a joint vision-language space at semantic level. Hence, we propose to leverage the image-text discrepancy to achieve universal fake image detection.

**Visual-language models.** Recent studies have demonstrated the great potential of vision-language models (VLMs) in learning general visual representation and aligning visual and text concepts Liu et al. (2024); Li et al. (2022a). The pre-trained VLMs have been proven to have impressive transferring ability to a variety of downstream tasks (Radford et al., 2021; Singh et al., 2022; Yuan et al., 2021). The CLIP model (Radford et al., 2021) could be a milestone of VLMs, as it employs transformer-based architecture (Dosovitskiy et al., 2021) with a contrastive pre-training strategy (Chen et al., 2020) for both image and text representation learning. There are already some works (Ojha et al., 2023; Cozzolino et al., 2023) that use pre-trained VLMs, such as CLIP, to learn image representation for detection. These methods, however, use only the visual space of VLMs, which could still lead to overfitting image patterns and cause insufficient learning without fully exploring VLMs' multi-modal potential. Whereas, we fully explore the multi-modal potential of VLMs by exploiting the distance between the images and corresponding captions in joint visual-language space at the semantic level, thus avoiding the overfitting of the visual-only image patterns and achieving improved generalization.

## 3 METHODOLOGY

### 3.1 IMAGE-TEXT DISCREPANCY REPRESENTATION

To exploit the discrepancy between image and text modalities, we first need to model the representation of these two modalities in a given visual-language latent space. CLIP (Radford et al., 2021) has been a milestone that optimizes an aligned vision-language space via contrastive learning. Hence, we propose to exploit the joint vision-language space of CLIP to learn the representation of image and text modality and explore their discrepancy.

For a given image $\mathbf{x}$ and its corresponding caption prompt $\mathbf{p}$, we feed the $(\mathbf{x}, \mathbf{p})$ into CLIP's image and text encoder, respectively, to obtain the visual and language representation $(\mathbf{I}, \mathbf{T})$, which can be formulated as follows:

$$(\mathbf{I}, \mathbf{T}) = \text{CLIP}(\mathbf{x}, \mathbf{p}), \tag{1}$$

where we use CLIP:ViT-L/14 with 768 output dimensions as our joint visual-language space.

Then we design a distance $\mathbf{D}$ to measure the discrepancy of $(\mathbf{I}, \mathbf{T})$ in the joint latent space. As the pre-training objective of CLIP is the cosine similarity between two modalities, we propose to use the subtraction of the two representations as their distance. The reason behind this design is that the subtraction of two representations is coherent with CLIP's objective, the cosine similarity, and our designed distance could provide higher dimensional information in latent space than one similarity score, such as direction, which should also contain informative clues for measuring discrepancy. We can formulate our image-text discrepancy representation as:

$$\mathbf{D} = |\mathbf{I} - \mathbf{T}|, \tag{2}$$

where $\mathbf{D}$ is our designed distance to measure the discrepancy of image and text modalities. Based on our observation in Fig. 1 (b), the $\mathbf{D}$ for fake images should be higher than the real images.

Thus, for a given image $\mathbf{x}$, we can measure its discrepancy distance $\mathbf{D}$ with the corresponding caption by first using a caption model to generate its caption $\mathbf{p}$. The distance serves as a clue for discriminating between fake and real images.

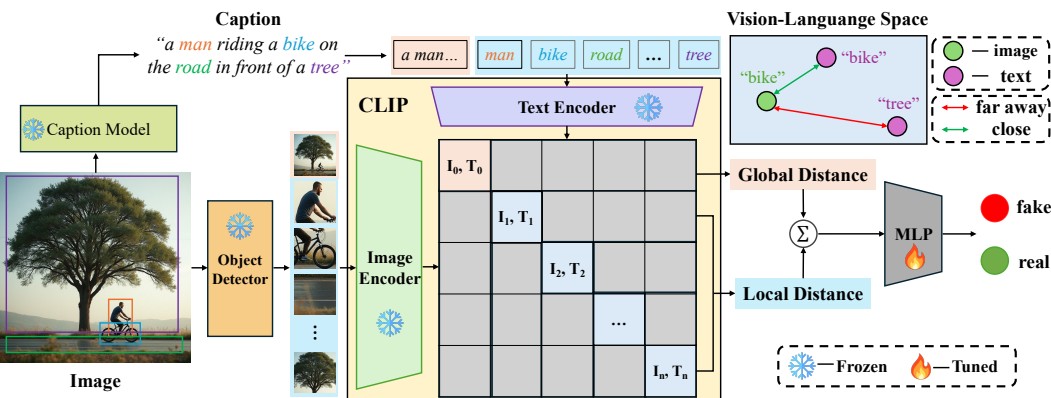

Figure 2: **Overview of our proposed method.** We explore the discrepancy between image and text modalities on both the global semantic clues, *i.e.,* the whole image, full caption, and local fine-grained semantic clues, *i.e.,* each local semantic object. After the representation learning stage, to perform the usual fake-image detection task, we optimize an MLP head.

## 3.2 GLOBAL-TO-LOCAL DISCREPANCY SCHEME

We have formulated the discrepancy between a given image $\mathbf{x}$ and the corresponding caption prompt $\mathbf{p}$ in a joint CLIP latent space. The discrepancy with the caption is mainly focused on the information of the whole image. This discrepancy, which we term a global discrepancy distance, could serve as a clue for discrimination. However, it ignores some more detailed fine-grained clues that can also contribute to detection. As shown in Fig. 1 (a), the forgery artifacts in fake images usually exist in local areas. The performance could be further boosted if we could explore more local fine-grained semantic clues. To this end, we further introduce a global-to-local discrepancy scheme to explore more fine-grained local semantic forgery clues, as illustrated in Fig. 2 and described as follows.

First, for a given image $\mathbf{x}$ and its corresponding caption $\mathbf{p}$, we denote the corresponding CLIP representation as $(\mathbf{I}_0, \mathbf{T}_0)$, and we define the discrepancy between the whole image and the full caption as global distance $\mathbf{D}_g$, formulated as:

$$\mathbf{D}_g = |\mathbf{I}_0 - \mathbf{T}_0|, \tag{3}$$

where $(\mathbf{I}_0, \mathbf{T}_0) = \text{CLIP}(\mathbf{x}, \mathbf{p})$. To further explore the fine-grained local semantic details, we focus on each semantic object described in the original full caption. We denote each semantic object with corresponding texts as $\{\mathbf{p}_1, \mathbf{p}_2, \cdots, \mathbf{p}_n\}$, and we employ a pre-trained object detection model to detect each corresponding semantic object in the original image as $\{\mathbf{x}_1, \mathbf{x}_2, \cdots, \mathbf{x}_n\}$ with their grounding boxes, as shown in Fig. 2. Then, we compute the discrepancy distance of each local semantic object formulated as:

$$\mathbf{D}_l^i = |\mathbf{I}_i - \mathbf{T}_i|, \quad i = 1, \cdots, n \tag{4}$$

where $(\mathbf{I}_i, \mathbf{T}_i) = \text{CLIP}(\mathbf{x}_i, \mathbf{p}_i)$ and the $\mathbf{D}_l^i$ is the local distance of $i$th semantic object. Then, we simply average all the local semantic objects as the final local distance:

$$\mathbf{D}_l = \frac{1}{n}\Sigma_{i=1}^n \mathbf{D}_l^i, \quad i = 1, \cdots, n \tag{5}$$

Finally, we obtain the distance that contains both global and local semantic clues by:

$$\mathbf{D} = w_1 \mathbf{D}_g + w_2 \mathbf{D}_l, \tag{6}$$

where the $\{w_1, w_2\}$ are the hyper-parameter weights for balancing the global and local distances.

## 3.3 FAKE-IMAGE DETECTION TASK

After the discrepancy representation learning stage, we obtain the desired distance representation that contains both global and local semantic clues. Then, we tune a classification head (a simple two-layer

MLP) to perform the usual fake image detection task by predicting the label based on input distance, which can be formulated as:

$$\hat{y} = \text{MLP}(\mathbf{D}, \theta_c), \tag{7}$$

where $\hat{y}$ is the predicted label and $\theta_c$ is the parameters of the MLP head. We employ a vanilla binary cross-entropy loss function to optimize the MLP, formulated as:

$$L(y, \hat{y}) = -\Sigma_{i=1}^{N} \left( y_i \log(\hat{y}_i) + (1 - y_i) \log(1 - \hat{y}_i) \right), \tag{8}$$

where $N$ is the mini-batch size, $y$ is the ground-truth label, and $\hat{y}$ is the corresponding prediction of the MLP head. Note that during training, only the MLP's parameters are optimized.

## 4 EXPERIMENT

### 4.1 EXPERIMENTAL SETUP

**Dataset.** Following recent works (Wang et al., 2020; Ojha et al., 2023), we first use the images generated by following models to evaluate our method, including seven different GANs: (1) Pro-GAN (Karras et al., 2018), (2) CycleGAN (Zhu et al., 2017), (3) BigGAN (Brock et al., 2018), (4) StyleGAN (Karras et al., 2019), (5) StyleGAN2 (Karras et al., 2020), (6) GauGAN (Park et al., 2019), and (7) StarGAN (Choi et al., 2018). We also follow four different diffusion models with various settings: (8) ADM (Dhariwal & Nichol, 2021), (9) LDM (Rombach et al., 2022), (10) Glide (Nichol et al., 2021), (11) DALLE (Ramesh et al., 2021), and one high-quality (12) deepfakes method[2]. To validate the performance on more recent and challenging generative models, we evaluate on recent DiffusionForensics (Wang et al., 2023) and GenImage (Zhu et al., 2024) dataset. As there exists an overlap between the two datasets, we choose ADM, Glide, and (13) VQDM (Gu et al., 2022) from GanImage, and (14) Stable-Diffusion-v1 (Rombach et al., 2022), (15) Stable-Diffusion-v2 (Rombach et al., 2022), LDM, (16) DALLE-2, (17) Midjourney, (18) ProjGAN (Sauer et al., 2021), StyleGAN, (19) Diff-ProjGAN (Wang et al., 2022), and 20) Diff-StyleGAN (Song et al., 2024) from Diffusion-Forensics dataset. Following prior works, we train our model and other baselines on the images generated by ProGAN from (Wang et al., 2020). To demonstrate our method does not highly rely on large-scale training data, we only use a subset that contains 4,0000 fake and real images, respectively.

**Evaluation metric.** Following prior state-of-the-art detectors (Wang et al., 2020; 2023; Ojha et al., 2023), we report accuracy (ACC) with a fixed 0.5 threshold and an average precision (AP) to evaluate our method and other baseline detectors.

**Baselines.** We compare our method with the following state-of-the-art baseline detectors: 1) ResNet-50 (He et al., 2016) with binary cross-entropy loss is a widely used backbone for image classification task. 2) Swin-Transformer (Liu et al., 2021) has a hierarchical transformer with shifted windows for downstream vision tasks. We use Swin-B/224×224 as our baseline. 3) Patchforensics (Chai et al., 2020) proposes a patch-wise classifier for detection at patch-level. 4) F3Net (Qian et al., 2020) proposes a two-stream network to mine two complementary frequency-aware clues. 5) DIRE (Wang et al., 2023) introduces a reconstruction error representation between the original and diffusion-reconstructed image to train the classifier. 6) CNNDet (Wang et al., 2020) carefully designs pre- and post-preprocessing and data augmentation to detect CNN-generated images. We use Blur+JPEG (0.1) setting as our baseline. 7) UniFD (Ojha et al., 2023) uses CLIP to extract only the image embeddings with the nearest neighbor as classification head. 8) NPR (Tan et al., 2024) explores the artifacts left by up-sampling layers in GAN and diffusion models to serve as discriminative clues. For a fair comparison, we use the same CLIP:ViT-L/14 for UniFD and our method. We train the aforementioned baselines from scratch with their released code using the same training set as ours. We categorize them into traditional image-classification backbones (ResNet-50 and Swin-T), deepfake detectors (Patchfor and F3Net), diffusion-generated image detectors (DIRE), and universal detectors (CNNDet, uniFD and NPR). Note that all the above baselines are visual-only detectors.

**Implementation details.** We use the pre-trained CLIP:ViT-L/14 to map the images and text prompts into 768 dimensions embeddings. The input images are center-cropped into $224 \times 224$, before being fed into CLIP. A simple fully-connected MLP is employed as our classification head, with an input dimension of 768 and an output dimension of 2, mapping the visual-language CLIP representation

---

[2]whichfaceisreal.com

Table 1: **Generalization results.** Accuracy (ACC) on CNNDetection and UniformerDiffusion datasets for detecting fake images from unknown generative models. Our method achieves an average improvement of 2.54% and 7.14% compared to recent UniFD and NPR.

| Detection method | Generative Adversarial Networks | | | | | | | Deepfakes | Diffusion Models | | | | | | | | Total |
| | Pro-GAN | Cycle-GAN | Big-GAN | Style-GAN | Style-GAN2 | Gau-GAN | Star-GAN | | ADM | LDM | | | Glide | | | DALLE | Avg. |
| | | | | | | | | | | 200 steps | 200 w/ CFG | 100 steps | 100 & 27 | 50 & 27 | 100 & 10 | | |
|---|---|---|---|---|---|---|---|---|---|---|---|---|---|---|---|---|---|
| ResNet-50 | 99.87 | 75.33 | 67.20 | 79.83 | 71.98 | 68.85 | 97.75 | 64.85 | 65.75 | 66.55 | 66.70 | 67.70 | 75.65 | 79.20 | 76.55 | 55.75 | 73.72 |
| Swin-T | 99.77 | 91.91 | 89.04 | 83.36 | 81.55 | 88.44 | 86.14 | 70.48 | 75.34 | 83.24 | 75.73 | 83.84 | 67.23 | 73.09 | 73.14 | 78.29 | 81.29 |
| Patchfor | 92.68 | 72.90 | 65.81 | 82.11 | 81.98 | 59.13 | 88.75 | 58.30 | 63.54 | 65.54 | 64.56 | 65.30 | 61.09 | 62.84 | 63.46 | 57.25 | 69.08 |
| F3Net | 99.85 | 71.56 | 77.54 | 90.46 | 80.72 | 60.28 | 99.79 | 54.88 | 64.93 | 77.44 | 76.59 | 77.29 | 84.29 | 86.14 | 85.59 | 75.09 | 78.90 |
| DIRE | 99.83 | 67.67 | 81.75 | 84.23 | 75.73 | 80.80 | 79.40 | 55.45 | 70.10 | 69.50 | 74.60 | 71.15 | 83.55 | 85.60 | 85.90 | 67.30 | 77.04 |
| CNNDet | 99.58 | 80.08 | 64.70 | 84.40 | 78.18 | 77.05 | 92.50 | 78.90 | 57.25 | 54.65 | 56.35 | 55.00 | 60.55 | 64.45 | 62.15 | 56.65 | 70.15 |
| UniFD | 99.65 | 93.00 | 95.70 | 85.85 | 75.55 | 99.45 | 95.30 | 81.55 | 75.20 | 94.05 | 78.45 | 94.15 | 79.65 | 81.70 | 79.25 | 86.20 | 87.17 |
| NPR | 99.90 | 77.58 | 78.90 | 93.30 | 96.43 | 75.20 | 99.60 | 64.55 | 74.70 | 81.70 | 82.40 | 82.55 | 81.45 | 83.55 | 85.45 | 63.85 | 82.57 |
| **TIDY** | **99.92** | **93.06** | 94.17 | 92.49 | 82.03 | 90.83 | 97.86 | **86.56** | **77.29** | 92.92 | 80.26 | 93.33 | 87.29 | 89.38 | 88.44 | 89.48 | 89.71 |

Table 2: **Generalization results.** Average precision (AP) on CNNDetection and UniformerDiffusion datasets for detecting fake images from unknown generative models. Our method achieves an average improvement of 1.69% and 8.02% compared to recent UniFD and NPR.

| Detection method | Generative Adversarial Networks | | | | | | | Deepfakes | Diffusion Models | | | | | | | | Total |
| | Pro-GAN | Cycle-GAN | Big-GAN | Style-GAN | Style-GAN2 | Gau-GAN | Star-GAN | | ADM | LDM | | | Glide | | | DALLE | mAP |
| | | | | | | | | | | 200 steps | 200 w/ CFG | 100 steps | 100 & 27 | 50 & 27 | 100 & 10 | | |
|---|---|---|---|---|---|---|---|---|---|---|---|---|---|---|---|---|---|
| ResNet-50 | 99.99 | 83.11 | 77.42 | 98.23 | 96.23 | 78.92 | 99.88 | 67.49 | 76.34 | 78.99 | 78.06 | 79.31 | 84.67 | 87.92 | 86.53 | 58.99 | 83.26 |
| Swin-T | 99.99 | 99.42 | 95.80 | 92.24 | 98.34 | 96.87 | 99.76 | 76.62 | 84.99 | 92.14 | 86.55 | 92.33 | 74.70 | 80.46 | 81.53 | 87.08 | 89.93 |
| Patchfor | 98.16 | 81.81 | 74.77 | 89.60 | 90.37 | 65.66 | 96.05 | 63.37 | 71.12 | 75.49 | 74.72 | 75.33 | 69.56 | 70.56 | 71.85 | 68.32 | 77.30 |
| F3Net | 99.99 | 79.20 | 89.83 | 99.03 | 99.02 | 66.86 | **100.0** | 58.16 | 75.00 | 87.92 | 84.17 | 87.46 | 92.39 | 93.89 | 93.44 | 84.99 | 86.96 |
| DIRE | 99.99 | 76.49 | 91.24 | 96.12 | 94.59 | 86.74 | 99.87 | 53.32 | 79.74 | 77.37 | 82.59 | 79.08 | 91.69 | 93.87 | 93.85 | 74.98 | 85.72 |
| CNNDet | 99.99 | 90.77 | 87.57 | 99.26 | 98.62 | 92.70 | 98.01 | **98.54** | 72.98 | 69.93 | 70.37 | 70.97 | 77.45 | 83.15 | 80.63 | 69.26 | 84.56 |
| UniFD | 99.99 | 99.77 | 98.90 | 98.19 | 97.64 | 99.94 | 99.62 | 96.99 | 87.00 | 97.14 | 89.11 | 96.98 | 89.69 | 91.02 | 89.65 | 93.76 | 95.34 |
| NPR | 100.0 | 97.28 | 86.93 | 98.98 | 99.42 | 78.85 | 100.0 | 61.04 | **88.31** | 89.61 | 90.03 | 90.14 | 89.02 | 90.78 | 92.01 | 71.77 | 89.01 |
| **TIDY** | 100.0 | 99.88 | 99.31 | 99.17 | 99.09 | 99.95 | 99.96 | 97.93 | 87.27 | 97.87 | 91.56 | 98.06 | 94.75 | 96.55 | 95.55 | 95.52 | 97.03 |

into real/fake predictions. To generate the caption of input images, we use BLIP-2 (blip2-opt-2.7b) (Li et al., 2023), and to detect each local semantic object, we use GLIP (glip-Swin-L) (Li et al., 2022c). We train the classification head by 50 epochs with vanilla binary cross-entropy loss. An Adamw (Loshchilov & Hutter, 2018) optimizer with $1e-3$ learning rate and $1e-3$ weight decay is employed to optimize the training process. We empirically set both the weights $\{w_1, w_2\}$ of global and local distance to 1.0. All experiments are conducted on NVIDIA A100.

## 4.2 COMPARISON TO STATE-OF-THE-ART

**Generalization to unknown models.** We begin by evaluating the detectors' generalization to unknown generative models, which is a critical challenge in this field. We train all the detectors with the same training dataset and then evaluate them on the aforementioned testing datasets. First, we evaluate them on the CNNDet (Wang et al., 2020) and UniformerDiffusion (Ojha et al., 2023) datasets, and the ACC/AP results are shown in Tab. 1&2. From the results, we observe that naive detectors, such as ResNet-50 and Swin-T, cannot achieve the desired performance on the unknown generative models. The detector designed for CNN-generated images, such as CNNDet, suffers from diffusion-generated images, as well as the detector designed for diffusion-generated images, such as DIRE, as it struggles to detect GAN-generated images. The universal detectors, including UniFD and NPR, achieve considerable performance on various unseen models. But, they still encounter slight performance drops on specific models, such as StyleGAN2 for UniFD and DALLE for NPR, which we assume is caused by the unseen model architectures or different image distributions. Whereas, our proposed method achieves impressive performance on various kinds of generative models, with an average improvement of 12.31% AP compared to DIRE, 1.69% compared to UniFD, and 8.02% compared to NPR. This provides evidence for our assumption that a universal detector should not rely on specific generated image data thus proving the superiority of our method by exploring the discrepancy in joint vision-language space.

To further support the impressive generalization of our method on various generative models, we conduct experiments on other recent DiffusionForensics and GenImage datasets. The ACC/AP results

Table 3: **Generalization results on more unknown models.** Detection accuracy and average precision (ACC/AP) averaged over real and fake images to more unknown diffusion models and generative adversarial networks from DiffusionForensics and GenImage datasets.

| Detection method | Diffusion Models | | | | | | | | Generative Adversarial Networks | | | | Total |
|---|---|---|---|---|---|---|---|---|---|---|---|---|---|
| | ADM | Glide | VQDM | SD-v1 | SD-v2 | LDM | DALLE-2 | Mid. | Proj-GAN | Style GAN | Diff-ProjGAN | Diff-StyleGAN | Avg. |
| ResNet-50 | 68.00/79.95 | 71.50/83.00 | 52.35/54.89 | 76.45/79.75 | 74.65/79.91 | 58.60/84.95 | 73.47/76.41 | 90.27/83.17 | 50.40/81.19 | 55.80/93.47 | 50.65/81.13 | 93.95/94.99 | 68.01/81.07 |
| Swin-T | 62.68/80.77 | 58.73/74.22 | 66.18/81.55 | 53.48/61.24 | 64.47/73.94 | 81.69/94.96 | 76.18/77.61 | 90.79/80.03 | 50.88/71.29 | 69.78/86.71 | 50.23/55.72 | 87.79/91.69 | 67.74/77.48 |
| Patchfor | 56.96/63.91 | 58.98/65.57 | 64.24/75.47 | 73.14/89.44 | 76.14/88.66 | 81.38/92.71 | 82.65/94.95 | **91.56**/97.97 | 63.86/80.11 | 64.57/80.10 | 64.54/79.85 | 84.89/94.60 | 71.91/83.61 |
| F3Net | 72.29/81.20 | 73.39/82.53 | 66.13/76.12 | 78.14/93.37 | **80.19**/89.11 | 87.89/97.40 | 90.79/96.81 | 87.29/95.28 | 72.49/96.97 | 88.10/95.35 | 65.98/95.71 | 92.49/99.25 | 79.60/91.59 |
| DIRE | 75.25/85.47 | **81.45**/90.49 | 66.85/76.79 | 74.05/81.80 | 73.10/88.97 | 80.65/**98.48** | 73.87/94.63 | 63.91/86.80 | 61.40/51.16 | 75.60/88.25 | 61.40/55.44 | 79.85/94.59 | 72.28/82.74 |
| CNNDet | 56.45/72.39 | 57.90/74.36 | 54.20/61.73 | 50.25/74.85 | 50.05/64.16 | 53.65/78.37 | 66.80/63.15 | 90.82/91.42 | 56.00/88.07 | 82.15/**98.65** | 55.25/85.57 | 95.35/**99.87** | 64.07/79.38 |
| UniFD | 73.15/85.62 | 61.95/72.57 | 84.30/93.07 | 74.20/93.91 | 65.25/**90.48** | 85.00/90.19 | **96.50**/98.71 | 73.00/95.44 | **88.80**/97.61 | 80.70/96.49 | 87.90/94.98 | 83.85/97.35 | 79.55/**92.20** |
| NPR | 70.80/81.72 | 71.95/88.82 | 67.80/71.91 | **81.25**/88.85 | 76.40/88.95 | 80.45/84.97 | 66.67/71.32 | 90.91/87.25 | 79.45/97.36 | 85.95/96.18 | 84.65/**99.02** | **95.75**/99.12 | 79.34/87.96 |
| **TIDY** | **87.24/93.00** | 79.11/90.59 | **86.93/93.87** | 80.52/**94.11** | 77.86/89.90 | **89.79**/97.58 | 91.19/**98.82** | 89.75/**98.08** | 92.29/97.99 | 89.01/96.55 | 88.13/96.03 | 95.52/99.34 | **87.29/95.49** |

are shown in Tab. 3. From these results, we can observe that our proposed universal detector achieves impressive generalization to more unknown GAN and diffusion models, when compared to other state-of-the-art competitors.

**Robustness to unseen perturbations.** The robustness to unseen perturbations is also a critical concern for current fake image detectors, as there are various post-preprocessing perturbations in real-scenario applications, such as compression. To address this issue, we evaluate all detectors' robustness against three common types of perturbations on images generated from ProGAN (the same as the training set), including Gaussian Noise, Gaussian Blur, and JPEG Compression following (Wang et al., 2020; 2023). For each perturbation, we consider three different severity levels: $\sigma = 0.001, 0.005, 0.01$ for Gaussian Noise, $\sigma = 1, 2, 3$ for Gaussian Blur, and $quality = 75, 50, 25$ for JPEG Compression. The results are shown in Fig. 3. We observe that our method suffers less from the three different perturbation types, with only a very slight performance drop compared to other baselines, especially under Gaussian Noise and Gaussian Blur. This indicates that leveraging the joint visual-language latent space of a pre-trained CLIP model can lead to a more robust representation for detecting fake images than using only visual image patterns.

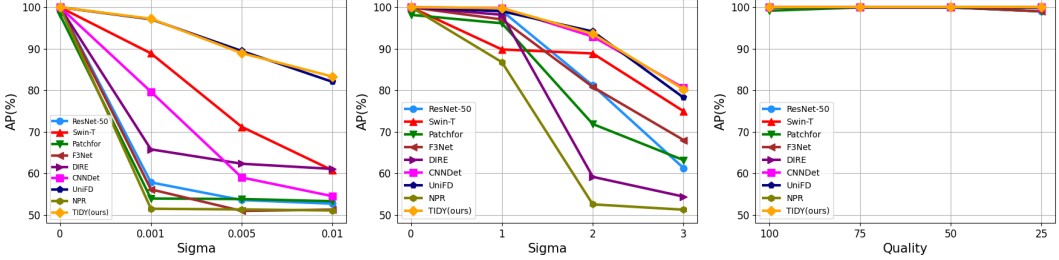

Figure 3: **Robustness results to unseen perturbations.** Average precision (AP) of different methods when detecting real/fake images under three different types of perturbations with three different severity levels: Gaussian Noise ($\sigma = 0.001, 0.005, 0.01$), Gaussian Blur ($\sigma = 1, 2, 3$), and JPEG Compression ($quality = 75, 50, 25$) (from left to right).

## 4.3 ABLATION STUDY

**Vision-language modalities vs. single modality.** To validate that multi-modalities could lead to improved detection compared to single modality, we first conduct an ablation study on different modalities of the training the detector . We use the following different variants: (i) only text, (ii) only image, and (iii) both (our TIDY). Note that, for both single-modality and multi-modalities settings, the embeddings used to train the MLP header are from the same CLIP architecture with 768 dimensions. The results are shown in Fig. 4. From the results, we observe that using only text performs worse than using only images, which could attribute to that all forgery clues or artifacts in images are ignored. Our method achieves improved performance compared to using only image or only text modality. This provides more evidence of our method's superiority, as it explores the vision-language discrepancy in a joint latent space instead of training on only a single modality.

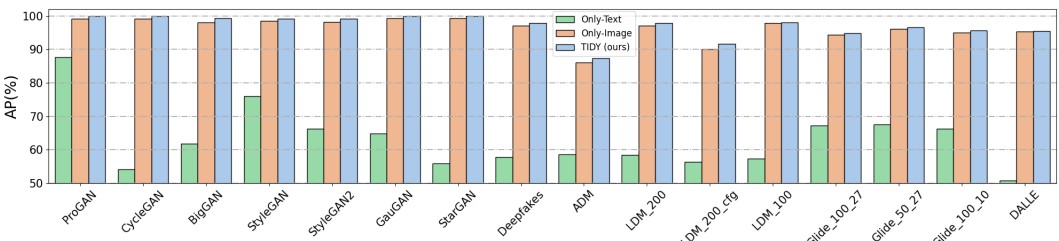

Figure 4: **Ablation study on different modalities.** The average precision (AP(%)) is reported. We observe that our method equipped with multi-modalities achieves improved performance compared to using only one single modality.

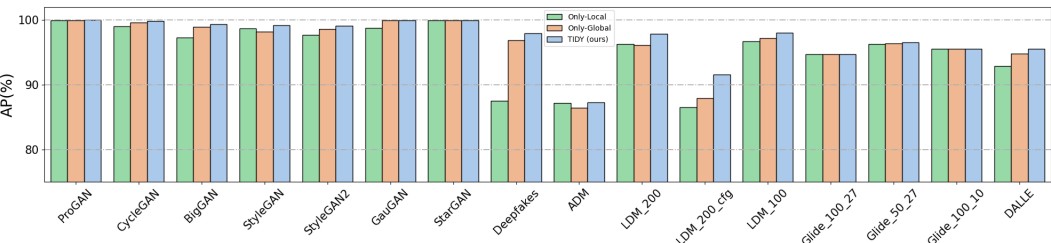

Figure 5: **Ablation study on different distances.** The average precision (AP(%)) is reported. We observe that our proposed local and global distances could both achieve impressive performance, and the performance is further boosted with equipped both.

**Effect of different distances.** To demonstrate that both our designed global and local distances contribute to the improved performance of the detection, we conduct an ablation study on our proposed global-to-local discrepancy scheme by employing the following variants: (i) only-local distance, (ii) only-global distance, and (iii) both distances. The results are shown in Fig 5. We observe that using only the local or global distance could achieve both an impressive performance, and the performance is further boosted when both are employed. The results support our hypothesis that the discrepancy exists in both the whole image and local areas. Exploring both global and more detailed fine-grained discrepancy clues leads to further improvements. This also provides more evidence of the effectiveness and superiority of our proposed method, as it uses the vision-language discrepancy from global to local perspective.

**Effect of different training datasets.** To evaluate whether our detector is universal when training data changes, we conduct experiments by using different generative models and image sources as the training set. We consider both the GAN and diffusion models. Specifically, we evaluate the following two variants: (i) ADM (Dhariwal & Nichol, 2021) trained on ImageNet (Russakovsky et al., 2015), and (ii) ProGAN (Karras et al., 2018) trained on LSUN. Note that, unless specifically stated ,the real images for training our detector are the same as when training the generative models. The results

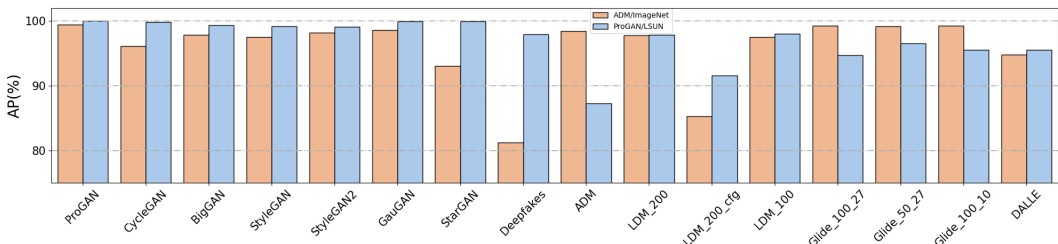

Figure 6: **Ablation study on different training datasets.** The average precision (AP(%)) is reported. Our proposed TIDY achieves impressive results, regardless of the generative models (e.g., GAN or diffusion models) and image source (e.g., ImageNet or LSUN).

are shown in Fig. 6. We observe that our method, when trained on diffusion-generated images can achieve impressive performance on GANs, and the same for detecting diffusion-generated images, when trained on GAN-generated images. Additionally, different training datasets can achieve similar impressive performance, irrespective of the generative models or image sources. This provides more evidence that our proposed detector is universal to unseen generative models, irrespective of different training datasets, *i.e.*, generative models or image sources.

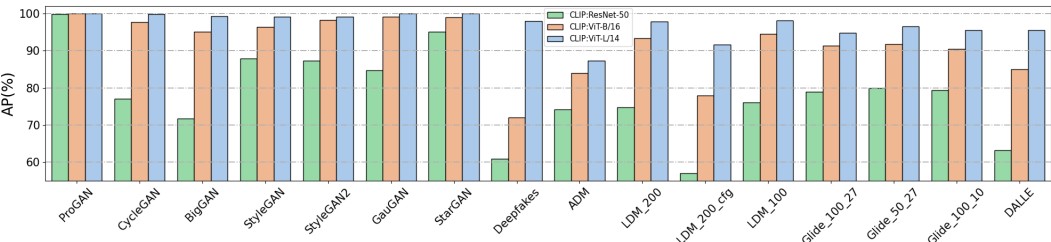

Figure 7: **Ablation study on different CLIP architectures.** The results indicate that the detection performance benefits from a larger CLIP backbone architecture.

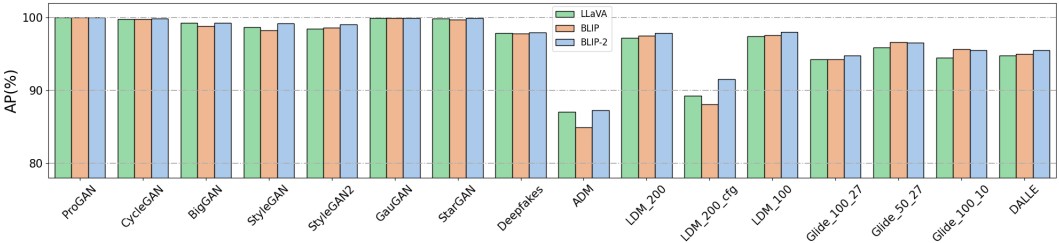

Figure 8: **Ablation study on different caption models.** Our method is robust to different caption models, which indicates that the discrepancy between image and caption is a general phenomenon.

**Effect of different CLIP architectures.** We conduct experiments to investigate the effect of different CLIP backbone architectures. We consider the following different architectures: (i) CLIP:ResNet-50, (ii) CLIP:ViT-B/16, and (iii) CLIP:ViT-L/14. We only change the CLIP architecture while keeping other settings the same and Fig. 7 shows the average precision of these variants to unseen generative models. From the results, we observe that variations in CLIP spaces could influence the performance. Specifically, the transformer-based CLIP architecture performs better than ResNet-50, which could be explained by its large-scale architecture and the long-range receptive field introduced by the attention blocks. ViT-L/14 also achieves higher performance than ViT-B/16, which could also be attributed to the larger backbone architecture.

**Effect of different caption models.** We conduct further ablations to demonstrate our method's effectiveness when employing different caption models. Specifically, we consider following different caption models: (i) LLaVA Liu et al. (2024), (ii) BLIP Li et al. (2022b), and (iii) BLIP-2 Li et al. (2023). Note that the prompt we use for LLaVA is: "Please generate a one-sentence caption for the input image." The results are shown in Fig. 8 and we observe that our method still achieves impressive performance when employing a different caption model, with only a slight drop compared to BLIP-2. This indicates that our observation and method are general and applicable to different caption models.

### 4.4 VISUALIZATION

To analyze whether our proposed representation could effectively distinguish the real and fake images in latent space, we visualize the distance representation by using t-SNE (Laurens & Hinton, 2008) on different models, including ProGAN (Karras et al., 2018) for a GAN model, LDM (Rombach et al., 2022) for a diffusion model, and StarGAN (Choi et al., 2018) for a generated face for deepfakes. The results are shown in Fig. 9. From the results, we first observe that our designed representations of real and fake images are clustered with a clear discrepancy margin in latent space for all three different generative models. This indicates that our representation has strong discriminability in distinguishing

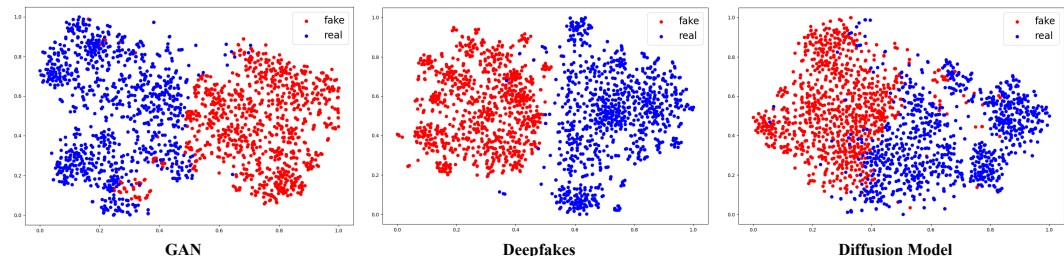

Figure 9: **The t-SNE visualization of our representation.** The real and fake images are clustered with a clear discrepancy margin in latent space, which indicates our representation preserves strong discriminability in distinguishing fake images from real ones.

between real and fake images. This provides more evidence about the effectiveness of our designed representation on the universal fake image detection task.

## 5    CONCLUSION

In this paper, we find that fake images exhibit significant discrepancies between images and corresponding captions compared to real images in joint visual-language space. Upon this observation, we reframe the fake image detection from a multimodal image-text perspective and propose **TIDY** to achieve universal fake image detection. Specifically, we first measure the distance between images and corresponding captions in a joint visual-language space of pre-trained CLIP and then tune an MLP head for detection. Considering the semantic divergence of local and global patches, and the artifacts in local patches of fake images, we further introduce a global-to-local discrepancy scheme to mine more fine-grained local semantic clues. Specifically, we propose to explore the discrepancy on the whole image and each semantic object described in the caption. Extensive experiments demonstrate our method's superiority against other state-of-the-art competitors in detecting various fake images with impressive generalization and robustness. We hope our method could provide insight on how to formulate the AI-generated image detection task from a multi-modal perspective and also foundations on how to leverage large pre-trained models to detect AI-generated content (AIGC) for future research. In the future, we aim to extend our idea and method to other AIGC detection tasks and facilitate the development of AIGC safety.

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
