APPENDIX

In this supplementary material, we provide more details about the baselines, related datasets, perturbations for robustness evaluation, and further ethical discussions about how our method could contribute to the community, described in detail as follows.

The code will be released at https://github.com/anonymous/TIDY_ICLR_3812.

## A  MORE DETAILS ON BASELINES

We present a brief description of state-of-the-art baselines for our comparisons in Sec. 4.1. Here, we summarize more details of each method of detector type, modality, and dependency for better comparisons, as shown in Tab. 4.

Table 4: Details of all baselines and our proposed method.

| Method | Detector Type | Modality | Clue |
|---|---|---|---|
| ResNet-50 (He et al., 2016) | Backbone | Image | image patterns |
| Swin-T (Liu et al., 2021) | Backbone | Image | image patterns |
| Patchfor (Chai et al., 2020) | Deepfakes | Image | local patch patterns |
| F3Net (Qian et al., 2020) | Deepfakes | Frequency | frequency patterns |
| DIRE (Wang et al., 2023) | Diffusion Models | Image | reconstruction error |
| CNNDet (Wang et al., 2020) | Universal | Image | pre- and post-preprocessing |
| UniFD (Ojha et al., 2023) | Universal | Image | pretrained visual space |
| NPR (Tan et al., 2024) | Universal | Image | up-sampling operation |
| TIDY (ours) | Universal | Image-Text | image and caption discrepancy |

## B  MORE DETAILS ABOUT THE DATASETS

In this section, we describe more details about the training and testing datasets we used for our experiments. As described in Section. 4.1, we choose totally 20 different generative models following recent (Wang et al., 2020; Ojha et al., 2023; Wang et al., 2023; Zhu et al., 2024). We categorize them into different generative model families, *e.g.*, GANs and diffusion models, with the training image source and resolution, as listed in Tab. 5 (the unconditional/conditional ADM are counted as one generative model).

Specifically, the training set includes 40,000 real and 40,000 fake images generated from ProGAN when trained on ImageNet (Russakovsky et al., 2015), and the test set of most generative model includes 1,000 real and 1,000 fake images (except DALLE2 includes 500 and Midjourney includes 100 fake with an equal number of real images). The resolution of most generated images is $256 \times 256$ (*e.g.*, ProGAN, StyleGAN, CycleGAN, GauGAN, *etc.*). For the images with higher resolution (*e.g.*, SD-v1, SD-v2, DALLE2, *etc.*), the generated images are resized into $256 \times 256$ with bicubic interpolation. Note that the real images are from the corresponding training set of each generative model unless specifically stated. Moreover, for better comprehension, we present examples from each generative model as shown in Fig. 10.

## C  EXAMPLES UNDER PERTURBATIONS

In Section. 4.2, we evaluate the robustness of our method under three different types of perturbations and make comparisons with other baselines. The three perturbations, in particular, include Gaussian Noise, Gaussian Blur, and JPEG Compression with three different severity levels. For better comprehension, we present some examples under each perturbation and severity level as shown in Fig. 11.

| Family | Generative Model | Image Source | Resolution |
|---|---|---|---|
| Unconditional GAN | ProGAN (Karras et al., 2018) | LSUN (Yu et al., 2015) | $256 \times 256$ |
| | StyleGAN (Karras et al., 2019) | LSUN (Yu et al., 2015) | $256 \times 256$ |
| | StyleGAN2 (Karras et al., 2020) | LSUN (Yu et al., 2015) | $256 \times 256$ |
| | BigGAN (Brock et al., 2018) | ImageNet (Russakovsky et al., 2015) | $256 \times 256$ |
| | ProjGAN (Sauer et al., 2021) | LSUN (Yu et al., 2015) | $256 \times 256$ |
| | Diff-ProjGAN (Wang et al., 2022) | LSUN (Yu et al., 2015) | $256 \times 256$ |
| | Diff-StyleGAN (Song et al., 2024) | LSUN (Yu et al., 2015) | $256 \times 256$ |
| Conditional GAN | CycleGAN (Zhu et al., 2017) | ImageNet (Russakovsky et al., 2015) | $256 \times 256$ |
| | GauGAN (Park et al., 2019) | COCO (Lin et al., 2014) | $256 \times 256$ |
| | StarGAN (Choi et al., 2018) | CelebA (Liu et al., 2015) | $256 \times 256$ |
| Deepfakes | WFIR (West & Bergstrom, 2023) | FFHQ (Karras et al., 2019) | $1024 \times 1024$ |
| Unconditional DM | ADM (Dhariwal & Nichol, 2021) | LSUN (Yu et al., 2015) | $256 \times 256$ |
| Conditional DM | ADM (Dhariwal & Nichol, 2021) | ImageNet (Russakovsky et al., 2015) | $256 \times 256$ |
| Text-to-Image DM | Glide (Nichol et al., 2021) | COCO (Lin et al., 2014) | $256 \times 256$ |
| | LDM (Rombach et al., 2022) | LAION (Schuhmann et al., 2021) | $256 \times 256$ |
| | SD-v1 (Rombach et al., 2022) | LSUN (Yu et al., 2015) | $512 \times 512$ |
| | SD-v2 (Rombach et al., 2022) | LSUN (Yu et al., 2015) | $768 \times 768$ |
| | VQDM (Gu et al., 2022) | ImageNet (Russakovsky et al., 2015) | $256 \times 256$ |
| | DALLE2 (Ramesh et al., 2022) | LSUN (Yu et al., 2015) | $1024 \times 1024$ |
| | Mid. (Midjourney, 2023) | LSUN (Yu et al., 2015) | $1024 \times 1024$ |

Table 5: **Details of the generative models for our evaluation**, including the model family, training image source, and resolution. We evaluate on various generative models, including GANs, diffusion models, and deepfakes.

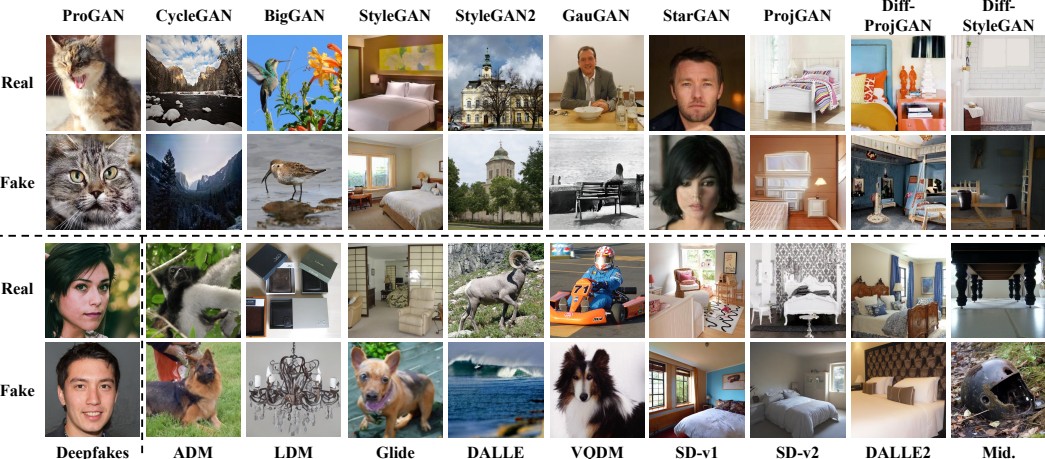

Figure 10: **Examples from different generative models**, including GANs, Deepfakes and Diffusion models from (Wang et al., 2020; Ojha et al., 2023; Wang et al., 2023; Zhu et al., 2024),

# D ETHICAL DISCUSSIONS

With the rapid development of current generative models, the competition between generation and detection is always in progress. Prior detectors may suffer from the upcoming generative models, and the new generative models can promote the design of new detectors. To achieve universal detection, our method leverages the discrepancy between the image and corresponding caption in a joint visual-language space; this discrepancy in generated images is general to various different generated images, including GANs, diffusion models, and deepfakes. If the generative models in the future can perfectly align the different modalities, which we assume should be difficult to achieve, the detectors based on multi-modalities could fail. Nevertheless, we believe our method can still provide

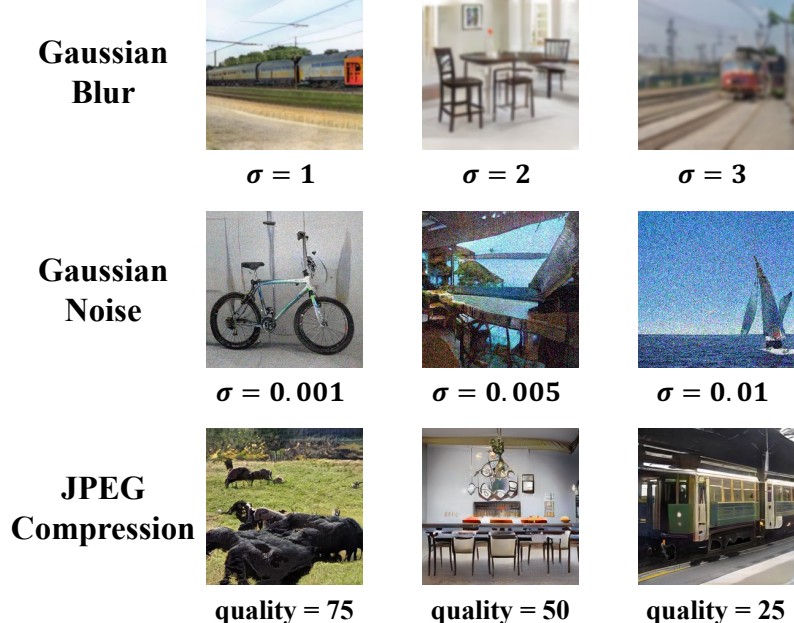

Figure 11: **Examples under three different perturbations** with three different severity levels, including Gaussian Blur, Gaussian Noise, and JPEG compression.

the foundation and insight into the related community about how to exploit the relationship between different modalities to achieve more general and robust detection.