# OpenReview forum: "Exploring Image-Text Discrepancy for Universal Fake Image Detection"
_ICLR.cc/2025/Conference — ICLR 2025 Conference Withdrawn Submission_

### Official Review · Reviewer_md9u · 2024-10-30

**Soundness:** 2
**Presentation:** 3
**Contribution:** 2
**Rating:** 5
**Confidence:** 4

**Summary:**

This paper introduces a universal fake image detection method called "TIDY," which leverages discrepancies between images and their corresponding text descriptions within a joint vision-language space to identify fake images. The study demonstrates that fake images exhibit greater discrepancies in image-text alignment compared to real images. The proposed method calculates the distance between images and text within the latent space of CLIP and further enhances detection by exploring both global and local discrepancies. Experimental results show that TIDY outperforms state-of-the-art methods in terms of generalization and robustness across various generative models

**Strengths:**

Good Global-to-Local Discrepancy Analysis: The proposed global-to-local discrepancy analysis enables fine-grained detection, addressing both overall image and localized semantic discrepancies, which enhances detection accuracy.
Extensive Experimental Validation: The authors conducted extensive experiments across diverse generative models, including GANs, diffusion models, and Deepfakes, demonstrating the method’s robustness and adaptability to new and unseen generative models.

**Weaknesses:**

The inference process for TIDY relies on both image and caption data, which introduces potential inefficiencies. Specifically, detection requires generating a caption for the manipulated image, processing it through a pre-trained model, identifying corresponding objects, and then computing the overall distance score between image and text. This reliance on caption generation and object detection may hinder the model’s efficiency and scalability in real-world scenarios where computational resources are limited or response time is critical. It would be beneficial for the authors to consider an image-only inference approach, which could streamline the process and make the model more lightweight and practical for broader applications without compromising detection accuracy.

After the object detector extracts objects, formula (4) calculates the distance between each object and its corresponding object token only, rather than between a single object and all object tokens. However, Figure 2 illustrates that the distance is large between an object and unrelated tokens. Could this description be inaccurate?

**Questions:**

See the weaknesses, my key concerns lie in the effiiency of proposed method and the complex pipeline, a sysmetic evaluation regarding the efficiency is required.

---

### Official Review · Reviewer_oYkK · 2024-11-02

**Soundness:** 2
**Presentation:** 2
**Contribution:** 2
**Rating:** 5
**Confidence:** 4

**Summary:**

The paper detects fake images from different generative models by leveraging image-text discrepancy between real and fake images. In light of this, authors explored the discrepancy on both global and local aspects. Experiments on two datasets show outstanding performance.

**Strengths:**

1.	Leveraging the cross-modal discrepancy in global and local is novel to some extent. And the method is simple yet effective.
2.	The paper is well-written and easy to follow.

**Weaknesses:**

1.	Lack comparison with recent works, such as FatFormer[1], FreqNet[2].
2.	This work shares a similar motivation with DE-FAKE[3]. The author is encouraged to provide a comparison with DE-FAKE.
3.	According to Figure 3, the performance under unseen perturbations is not outstanding. The author is suggested to provide a discussion.

[1] Forgery-aware Adaptive Transformer for Generalizable Synthetic Image Detection. CVPR 2024
[2] Frequency-Aware Deepfake Detection: Improving Generalizability through Frequency Space Learning. AAAI 2024
[3] DE-FAKE: Detection and Attribution of Fake Images Generated by Text-to-Image Generation Models. ACM CCS 2023

**Questions:**

1.	Would the object detection performance affect subsequent learning? How to avoid the mismatch between “object patch” and text embeddings?
2.	Why the fake images show more distinct discrepancies with the caption than real images. The author is encouraged to provide some discussion in the paper.

---

### Official Review · Reviewer_CfDE · 2024-11-04

**Soundness:** 2
**Presentation:** 2
**Contribution:** 1
**Rating:** 3
**Confidence:** 5

**Summary:**

The paper introduces a method that detects AI-generated fake images by leveraging discrepancies between images and their corresponding textual descriptions in a shared vision-language embedding space using CLIP. Unlike traditional detectors that focus solely on visual features and may overfit to specific artifacts, this approach captures differences more pronounced in fake images by analyzing both global image-level discrepancies and local discrepancies at the level of individual semantic objects. By combining these discrepancies, the method trains a simple MLP classifier to distinguish between real and fake images. Extensive experiments demonstrate that this method outperforms some other detectors in accuracy, and generalization to unseen generative models.

**Strengths:**

- This paper  introduces a unique method that exploits image-text discrepancies in a joint embedding space, moving beyond traditional visual-only detection methods.
- The inclusion of both global and local discrepancies allows for capturing fine-grained forgery clues.

**Weaknesses:**

- The method relies on captions for images. Inaccurate or misleading captions could negatively impact detection performance, especially for complex scenes or low-quality images.
- The approach assumes that a caption can be accessed for every image, which may not hold in all cases, particularly for images with abstract content or minimal semantic information.
- Errors in object detection and caption generation stages can propagate and affect the final detection performance.
- The method is only evaluated on out-of-date models, results on more challenge models like FLUX, SD3, DALLE3 etc will make it more convincing.

**Questions:**

See weaknesses above.
How to detect fake images without any caption?

---

### Official Review · Reviewer_VzPG · 2024-11-12

**Soundness:** 3
**Presentation:** 3
**Contribution:** 2
**Rating:** 3
**Confidence:** 5

**Summary:**

This paper proposes to tackle the fake image detection problem via utilizing the discrepancy in the vision-language space (based on pre-trained CLIP model), in which fake image-text pairs exhibit higher discrepancies than the real ones thus helping the discrimination among them. Basically, given an image to be tested, the off-the-shelf image captioning model is adopted to first generate the corresponding text description of the input image, followed by feeding the image-text pair (as well as the object proposals found in the image and the objects described in the text) to the CLIP model for computing the discrepancy (between vision and language representations). The proposed method demonstrates better performance than baselines, verifying its efficacy and generalizability.

**Strengths:**

+ The proposed method (named as TIDY) experimentally shown to have superior performance and better generalization ability (across various datasets produced by face-swap, GAN-based generative models, and the diffusion-based generative models) with respect to several baselines/SOTAs.
+ The thorough ablation studies are provided to better verify the design choices and the properties of the proposed method.
+ The organization and writing of this paper is clear and easy to follow.

**Weaknesses:**

- The novelty of the proposed method seems to be limited:
1. The utilization of the joint representation (e.g. CLIP-based) over vision and language domains has been proposed in [Sha et al., DE-FAKE: Detection and Attribution of Fake Images Generated by Text-to-Image Generation Models, CCS'23], in which their method (implicitly) also leverages the discrepancy in the vision-language space to perform the deepfake detection. However, such important work is not included in this submission.
2. Furthermore, although the proposed method in this submission also considers the local discrepancy (in which the the object proposals found in the image and the objects described in the text are used), such design does not significantly boost the overall performance (according to the ablation study) and it needs to additionally utilize the object detector (where such object detector could be also limited to the object classes it has learn).
- The image-only model variant (cf. Figure.4) shows comparable performance with respect to the full model (and that is why the proposed method has only slightly better performance than the UniFD baseline, and such performance improvement could even disappear when the local discrepancy is excluded), there hence should be more investigation into better leveraging the complementary properties across vision and language domains.
-

**Questions:**

The authors should carefully address the concerns as listed in the weaknesses (e.g. limited novelty, insignificant performance improvement with respect to the baselines).

---

### Note · Authors · 2024-11-15

I have read and agree with the venue's withdrawal policy on behalf of myself and my co-authors.